# Exposure to Secondhand Smoke: Inconsistency between Self-Response and Urine Cotinine Biomarker Based on Korean National Data during 2009–2018

**DOI:** 10.3390/ijerph18179284

**Published:** 2021-09-02

**Authors:** Boram Sim, Myung-Bae Park

**Affiliations:** 1Health Insurance Review and Assessment Service (HIRA), Wonju 26465, Korea; cici1011@naver.com; 2Department of Gerontology Health and Welfare, Pai Chai University, Daejeon 35345, Korea

**Keywords:** secondhand smoke, thirdhand smoke, biomarker, cotinine, nonsmokers, Korea

## Abstract

This study aimed to estimate the secondhand smoke (SHS) exposure using urinary cotinine (UCo) to prove that the SHS exposure could not be properly assessed by self-reporting (SR). In total, 28,574 nonsmokers aged >19 years were selected from the Korean National Health and Nutrition Examination Survey data (2009–2018). First, changes in the annual concentration of UCo were analyzed, and the annual SHS exposure rates were measured based on SR and UCo from 2009 to 2018. Then, the average UCo concentration and UCo-measured SHS exposure rate were confirmed according to the subjects’ characteristics. Finally, factors associated with the UCo-measured SHS exposure rate were identified based on multiple regression analysis. The findings showed that the annual UCo concentrations and self-reported SHS exposure rates dropped significantly over the past decade. In contrast, the UCo-measured SHS exposure rate indicated that >80% of nonsmokers are still exposed to SHS. Moreover, we found vulnerable groups using UCo-measured SHS exposure rate. In particular, the self-reported SHS exposure at home and in workplaces and house type was highly associated with SHS exposure. Thus, these findings indicate that the actual SHS exposure could not be properly assessed by SR and should be verified using a biomarker, such as UCo. Considering that even a short-term exposure can be harmful to health, the goal of the policy should be to keep cotinine concentration as low as possible.

## 1. Introduction

Secondhand smoke (SHS) is composed of side stream smoke from burning tobacco or mainstream smoke that is exhaled by smokers. SHS contains > 7000 chemicals comprising hundreds of toxins and 70 types of carcinogens [1]. Thus, SHS is detrimental to nonsmokers’ health, particularly children and pregnant women. In children, SHS exposure may cause health problems such as severe asthma, respiratory infections, ear infections, and even sudden infant death syndrome, and in adults, it can cause stroke, lung cancer, and coronary heart disease. Approximately 600,000 nonsmokers die from SHS annually worldwide [2], accounting for 1% of the global disease burden and 10–15% of the disease burden caused by active smoking. SHS can also incur social costs, including direct healthcare expenditures as well as indirect costs, such as reduced productivity [3]. Of note, SHS can be present in any environment accessible to the public, including the workplace, bar, public transportation, or even at home [4]. The World Health Organization (WHO) estimated that 40% of children, 33% of adult male nonsmokers, and 35% of adult female nonsmokers are exposed to SHS worldwide [2].

Thus, the SHS exposure rate is globally used as an indicator of health behavior to protect the public from the risk of SHS. To date, several measurements have been proposed to accurately evaluate SHS exposure. The most common measurements include the self-reporting (SR) of one’s experiences of SHS exposure and biomarkers, including the presence of cotinine in serum, urine, and saliva. Cotinine is a metabolite of nicotine with a long half-life of 20–30 h. Among the biomarkers of SHS exposure, including carbon monoxide, carboxyhemoglobin, nicotine, cotinine, tobacco-specific carcinogen, and carcinogenic DNA abduct, cotinine has been widely used in the clinical field owing to its biological stability [5]. In each country, governments have developed a national survey with different measurements to monitor the health effects of SHS or the burden of disease due to SHS and to set the effectiveness of tobacco control programs or interventions [6]. For example, in the United States, SHS exposure rates are monitored by the Centers for Disease Control and Prevention using serum cotinine data collected from the National Health and Nutrition Examination Survey (NHANES). This rate decreased significantly from 87.5% to 25.2% between 1988 and 2014, but one in four individuals are still exposed to SHS [7]. In the United Kingdom, the National Health Service continues to monitor SHS exposure rates through SR and saliva, which are collected annually by the Health Survey for England. As of 2017, the self-reported exposure rates for SHS in the UK were 28% and 24% for men and women, respectively.

In South Korea, SR and urine tests cotinine (UCo) are collected through the Korean NHANES (KNHANES). However, when developing a national health plan, only the measurements of SR are officially reflected [8]. The SR results showed that exposure to SHS by adult nonsmokers in the workplace decreased from 49.2% in 2010 to 11.9% in 2018 and that the exposure rate at home also declined from 14.9% in 2010 to 4.7% in 2018 [9]. However, a problem that persists is that estimates measured using SR may differ from the ones measured using urinary cotinine (UCo). Several studies report the limitations of SR, including subjectivity, inaccurate responses, and the possibility of underestimating SHS exposure [10,11]. For example, some people may be more insensitive or tolerant to smoke exposure, resulting in an inaccurate report that they were not exposed to SHS [12]. However, even at very low concentrations of smoke, the risk of diseases, such as coronary heart disease and stroke, can significantly increase, and the Surgeon General’s report in the United States emphasized that there is no risk-free level of SHS exposure [5,13]. Thus, the government aims to set a goal to entirely eliminate SHS exposure to protect nonsmokers, even when they are unable to recognize their own exposure. In this sense, a biomarker, which quantifies the biologically active dose of nicotine and toxins in the body, can be a good subsidiary indicator, reflecting SHS exposure in all places. Thus, several attempts have been made to measure SHS exposure using UCo in Korea. Park et al. [14] confirmed the annual trend of SHS exposure using UCo concentrations as an SHS exposure rate indicator. Jung et al. [15] reported that a certain level of UCo was detected in nonsmokers who self-reported with no exposure experience. Moreover, Jeong et al. [12] confirmed that correspondence between the self-reported and UCo-measured SHS exposure rates were low. Cumulatively, these studies suggest that the actual SHS exposure is not adequately assessed by SR and needs to be verified using a biomarker, such as UCo.

Therefore, this study aimed to determine SHS exposure using UCo and assess its importance as a biomarker. Therefore, we analyzed UCo concentration and the SHS exposure rate with SR and UCo over a period of 10 years. In addition, UCo concentrations and the SHS exposure rate were confirmed using subject characteristics. Finally, factors associated with UCo-measured SHS exposure were identified to provide evidence on tobacco control policy.

## 2. Materials and Methods

### 2.1. Korean NHANES (KNHANES)

This study used data from the KNHANES, which is annually conducted by the Korea Centers for Disease Control and Prevention Agency (KCDA) since 1998. As a national statistic representing the whole Korean population, this survey selects approximately 4000 households and 10,000 individuals through stratified multistage clustered probability sampling according to age, sex, and region [9]. The survey collects an individual’s information on health behavior and socioeconomic characteristics using a structured questionnaire. The examination of health status using the laboratory testing of urine and blood is also conducted. For these tests, surveyors, nurses, and other personnel are given training on KNHANES. Under the standardized system on data management, surveyor codes data using computer-assisted personal interviewing and health examination results, including urine tests, are stored and managed by electronic health records in KCDA. Data are available in a year after the data cleaning processes for reviewing logical errors.

### 2.2. Subject Selection

We obtained 28,574 subjects’ data from KNHANES (data from 2009 to 2018, excluding 2012 and 2013). In the excluded years, the survey discontinued collecting UCo data temporarily. Among these data, we selected nonsmokers aged >19 years for this study. To exclude smokers, we used a questionnaire to assess smoking status. Ultimately, we selected 28,574 subjects for the study among 64,915 adults aged >19 years, excluding 17,008 cigarette or e-cigarette smokers (self-reported) and 19,333 subjects without UCo data.

Furthermore, when estimating the SHS exposure rate and to reduce the misclassification of self-reported smoking status, we verified smoking status once more using the subjects’ UCo concentrations. Based on a previous study conducted in Korea [16], we applied a cutoff point (UCo ≤ 20 ng/mL), which is an upper limit for distinguishing nonsmokers from active smokers. According to this criterion, 2147 of the total subjects were further classified as smokers, although they answered that they do not smoke (Figure 1).

### 2.3. Measurement of SHS Exposure Rate

We estimated the SHS exposure rate using two parameters: SR and biomarker, urinary cotinine (UCo). Firstly, in the SR measurement, we used questions on the SHS exposure experiences in three different places: home, public places, and workplaces. Respondents who answered “Yes” to the question “Have you ever smelled smoke of other people’s cigarettes in the indoor home in the past week?” are considered to have indoor home SHS exposure. Similar to an indoor home, the survey asks the same question concerning indoor workplaces and public places. Accordingly, we obtained SR SHS exposure rates in the home, workplaces, and public places.

Secondly, we estimated cotinine-based SHS exposure rate. Scientifically, if a person’s cotinine concentration is above the limit of detection (LOD), the person is considered to be in the SHS exposure group. In KNHANES, the LOD of UCo is <0.27399 ng/mL [17]. Subjects with UCo value greater than the cutoff point are classified as smokers and excluded from the measurements.

### 2.4. Covariates

This study includes factors known to be associated with SHS in previous studies: demographic and socioeconomic characteristics, environmental factors, health behavior, and self-reported SHS exposures. In general, women are more likely to be a nonsmoker exposed to SHS than men because smoking rates are higher in men than in women. However, it differs in terms of the location of exposure; in particular, women are more vulnerable to exposure at home, whereas men are more vulnerable to exposure at work [18]. Furthermore, children and adolescents who cannot control the behavior of smokers or cannot leave the exposed environment have a high risk of SHS exposure [19]. Low socioeconomic levels, including income levels, are also important factors affecting SHS exposure [20], and the SHS exposure also differs depending on the occupational characteristics [21]. Regarding environmental risk factors, rural areas have fewer legislations to ban smoking than urban areas; thus, the former have higher exposure rates than the latter [22]. Moreover, people living in multiunit housing, where smoke can seep into the individual’s living space from other houses or public spaces, are more vulnerable than those living in an ordinary house [23]. Drinking also strongly correlates with SHS as well as smoking [24,25]. Furthermore, cotinine levels may vary depending on SHS exposure experiences at home, indoor public places, and indoor workplaces measured by SR.

Therefore, we chose the following as control variables: sex, age, income, occupation, region, house type, monthly drinking, and self-reported SHS exposure at home, indoor public places, and indoor workplaces. We divided sex into male and female and categorized participants into 19–44, 45–64, and ≥65 age groups. We also established four income groups according to the household income quartiles. We classified occupational status into the following four categories: (i) blue-collar, (ii) white-collar, (iii) agriculture, and (iv) unemployed. Regions in which respondents lived were classified as urban and rural areas, and house types were categorized into general housing and apartment. We defined monthly drinking as the percentage of people who drank more than once a month in the last year. Meanwhile, self-reported SHS exposure was defined as the percentage of persons who smelled other people’s smoke at home, indoor public places, and indoor workplaces over the last 7 days.

### 2.5. Statistical Analysis

The changes in annual concentration of UCo were analyzed using the geometric mean and the changes in annual SHS exposure rates were measured using SR and UCo from 2009 to 2018. Then, the average UCo concentration and UCo-measured SHS exposure rates were confirmed according to the subjects’ characteristics. Finally, we identified factors related to the UCo-measured SHS exposure rate through univariate and multiple logistic regression analyses. All analyses were weighted according to stratified multistage sampling and additionally reflected the weights of 10 years of data integration.

## 3. Results

### 3.1. UCo Concentration and SHS Exposure Rate in Nonsmokers by Years

The subjects’ UCo concentration was 1.06 ng/mL in 2018 and has steadily declined from 9.28 ng/mL in 2009 to 1.03 ng/mL in 2016 but slightly increased to 1.06 ng/mL in 2017 and 2018. The self-reported SHS exposure rate in 2018 at three different places is as follows: 3.7% at indoor home, 15.4% at indoor public places, and 12.2% at indoor workplaces. Although there have been differences in the estimates depending on the location, the values have significantly decreased in all areas since 2009 and 2014. Finally, to determine the SHS exposure rate using UCo, we excluded subjects with UCo greater than the cutoff point (UCo > 20 ng/mL). The SHS exposure rate increased over the years until it peaked at 95.9% in 2014 and then declined but increased slightly again to 80.0% in 2018. For all years, the SHS exposure rate determined using UCo was higher than that measured using SR (Table 1).

### 3.2. UCo Concentration in Nonsmokers by Subjects’ Characteristics

Men had higher UCo concentration levels (2.0 ng/mL) than women (1.7 ng/mL), and the 19–44 age group showed the highest UCo level (2.2 ng/mL) among all age groups. The lower-income quartile had a higher UCo level, which was 2.0 ng/mL. The agriculture group had the highest UCo level (2.1 ng/mL) among all the occupational groups. Those who lived in the rural areas had a higher UCo level (2.0 ng/mL) than those who lived in urban areas (1.8 ng/mL). General housing residents had a higher UCo level (2.1 ng/mL) than apartment residents (1.6 ng/mL). Those who drank more than once a month (monthly drinkers) had a higher UCo level (2.1 ng/mL) than those who did not (1.5 ng/mL). Lastly, those who self-reported SHS exposure at home, indoor public places, and indoor workplaces had higher UCo levels (4.4, 1.7, and 3.5 ng/mL, respectively) than those who did not (Table 2).

### 3.3. UCo-Measured SHS Exposure Rate by Subjects’ Characteristics

Men had a higher SHS exposure rate (86.4%) than women (80.0%). The SHS exposure in the 45–64 age group was the highest (84.4%) followed by that in the 19–44 age group (82.5%) and the ≥65 age group (78.5%). The 1st quartile, which is the lowest income group, had the highest exposure rate of 83.7%, followed by the 3rd quartile with 83.2%, the 2nd quartile with 81.8%, and the 4th quartile with 81.5%. The SHS exposure rate by occupation was highest in the blue-collar group (87.9%), followed by white-collar (83.9%), agriculture (81.2%), and unemployment groups (79.2%). Regarding environmental characteristics, the SHS exposure rates were higher in urban area residents (82.8%) than in rural area residents (80.9%), and general housing residents had a higher exposure rate (85.9%) than those residing in an apartment (79.5%). A higher rate of SHS exposure was also found in those who drank once a month (84.9%) than in those who did not (80.0%). The SHS exposure rates of those who self-reported SHS exposure at home, indoor public places, and indoor workplaces were 93.0%, 89.3%, and 91.7%, respectively, which were higher than those who did not report SHS exposure (Table 3).

### 3.4. Factors Related to SHS Exposure Rates

Univariate logistic analysis revealed that only the incomes of 3Q and 2Q were not significantly different from those of 4Q. However, there were significant differences in all other variables. Factors related to SHS exposure rates were then identified by multiple logistic regression analysis. Men were 1.43 times more likely to be exposed to SHS than women. By age, the 19–44 age group was 1.21 times more likely to be exposed to SHS than the ≥65 age group. Although differences between income groups were not statistically significant, the occupation was significant. The white-collar (0.77 times) and agriculture (0.70 times) groups were less likely to be exposed to SHS than the blue-collar group. However, the blue-collar group showed no significant difference compared with the unemployment group. Moreover, two environmental characteristics were significant: urban area residents were 1.29 times more likely to be exposed to SHS than rural area residents, and general housing residents were 1.55 times more likely to be exposed to SHS than apartment residents. Furthermore, monthly drinkers were 1.17 times more likely to be exposed to SHS. Lastly, the self-reported SHS exposures at home, indoor public places, and indoor workplaces were all associated with a high UCo-measured SHS exposure rate. Those who self-reported SHS exposure at home, indoor public places, and indoor workplaces were 3.90, 1.51, and 2.06 times more likely to be exposed to SHS, respectively (Table 4).

## 4. Discussion

### 4.1. UCo Concentrations and SHS Exposure Rate

The primary novel finding of this investigation is that UCo concentrations significantly dropped over the past decade, possibly reflecting the results of the tobacco control policies. Considering the risks associated with smoking, policies have been put in place globally to regulate tobacco under the WHO framework Convention on Tobacco Control. Article 8 of this framework requires “the adoption of effective measures to protect people from exposure to tobacco smoke.” Accordingly, the smoke-free law, which restricts smoking in workplaces and communities, has expanded and contributed to the reduction in SHS exposure and SHS-induced health problems, including cardiovascular diseases and premature birth [26]. Moreover, although the law is supposed to protect nonsmokers from SHS, it helps smokers quit smoking and even prevents the establishment of the habit in the first place [27,28]. In this context, for the past 20 years, South Korea has committed to protect the public from smoking by gradually expanding smoke-free areas under the National Health Promotion Act, which designated all enclosed places as smoke-free areas in 2015 [8,12]. Public campaigns to raise awareness on the risk of smoking have been implemented through mass media, advertising on buses and within the subway, information dissemination over the internet, and other public programs, thereby promoting smoking cessation [29]. Consequently, smokers are becoming more aware of the risks of SHS exposure and have positively changed their attitudes toward antismoking laws. Moreover, these policies have had a positive effect on creating smoke-free rules at home [12]. The annual trends of self-reported SHS exposures rate in our study possibly reflect this positive change. However, the problem is that the UCo-measured SHS exposure rate remained high and gradually increased from 75.1% in 2009 to 80.0 in 2018. In other words, despite positive changes in UCo concentrations, 80.0% of people remained exposed to SHS at significant levels. The declining self-reported SHS exposures imply that the current tobacco control policy may have been more effective for highly exposed groups [8]. Therefore, a paradigm shift in tobacco control policies is required to further protect nonsmokers exposed to low concentrations of smoke. Considering that even short exposure to smoke can be harmful to health, the goal of the policy should be to reduce cotinine concentration to at least under the LOD level [7,23]. In the US, during 1988–2000, the laboratory LOD for serum cotinine was 0.05 ng/mL, but since 2001, the LOD decreased to 0.015 ng/mL. However, the CDC still uses LOD < 0.05 for historical comparisons [7,23]. In other words, reducing below the LOD is a policy level, and since there is no scientifically safe exposure level [30]. Ideally, it should aim to minimize the cotinine concentration of the population

### 4.2. Inconsistency between Self-Reported and UCo-Measured SHS Exposure Rate

This investigation reveals inconsistencies between self-reported and UCo-based SHS exposure rates. The UCo-measured SHS exposure rate was 80.0% in 2018 and approximately 5–21 times higher than those measured using SR, which were 3.7% for home, 15.4% for indoor public places, and 12.2% for indoor workplaces in 2018. This result is consistent with that in a previous study [10], which reported that cotinine-based measurements demonstrated higher SHS exposure rates than SR. Differences in exposure rates determined using cotinine and SR can be explained by the following points. First, SR merely relies on sight and smell. Hence, it is difficult to accurately assess whether smoke exists, given that 85% of the smoke is invisible [31]. Especially, SHS exposure from e-cigarettes is less recognizable [32]. Second, SR does not reflect thirdhand smoke (THS), which is adsorbed on various areas, such as walls, clothes, and carpets, and can be released back into the atmosphere again. Third, depending on social tolerance level, the degree to which an individual perceives SHS exposure may differ [33]. Last, the results from SR do not reflect various aspects of the situation during exposure; these aspects include the duration of exposure, number of smokers, size of the place, and amount of ventilation [11]. SR is advantageous because it provides specific information on SHS exposure at each place. However, the effects of SHS on the human body are not exactly clear simply from the investigation of whether it has been exposed to SHS in each place. For example, SHS exposure in public places and workplaces is more likely to occur for a relatively short period, whereas at home, it may last for a long time [34]. In other words, the proportion of people who responded that they were exposed to SHS among all respondents does not reveal the intensity of SHS exposure. To complement these limitations, this study employed a biomarker (cotinine)-based measurement. Biomarkers can address under-reporting issues for specific population groups that can occur in SR [35] and improve the quality of SHS measurements [8]. Biomarkers also allow the objective estimation of SHS exposure at all locations by measuring the accumulated nicotine and toxic substances in the body. Among biomarkers, cotinine is widely acknowledged as a valid indicator of SHS exposure owing to its biological stability. It has been used to provide evidence in previous studies that confirmed the causal relationship between SHS and SHS-induced diseases, including lung cancer and cardiovascular diseases [36,37].

### 4.3. Factors Associated with SHS Exposure Measured Using Urine Cotinine Level

In previous studies that measured SHS exposure rate based on SR using the KNHANES data, the younger-age group had a higher exposure risk. In addition, the low- and high-income groups were more likely to be exposed to SHS at home and in workplaces, respectively [24]. In terms of sex, women at home and men in workplaces were more likely to be exposed to SHS and those living in urban areas had a higher exposure risk than those in rural areas [24]. However, our study, which used biochemical indicators, demonstrated slightly different results compared with previous studies. Overall, our study showed that males, young adults, blue-collar workers, urban residents, general housing residents, and monthly drinkers had a higher risk of smoking exposure. Moreover, those who self-reported SHS exposure at home, inside public places, and indoor workplaces not only had high UCo levels but also had high UCo-measured SHS exposure rates.

Among the places of exposure, self-reported SHS exposure at home was most relevant to the exposure rate measured by UCo. According to studies in the United States and Spain, which measured SHS exposure by SR and cotinine, self-reported exposure at home has a greater impact on cotinine levels than exposure at work or elsewhere [38]. Similar results were found in the study that assessed the impact of SHS exposure on lung function using the same data source as that used in our study [39]. These results support previous studies suggesting that home is the primary source of SHS exposure [40], considering that SHS at home is highly concentrated or has long-term effects compared with SHS in other places. Koreans stay at home for a mean of 14.59 h/day [34]. Similarly, our study revealed that home SHS exposure by SR had the greatest effect on the SHS exposure rate by cotinine (3.9 times greater than non-SHS exposure group by SR). Therefore, an antismoking environment at home is crucial, although the self-reported SHS exposure rate at home was lower than that in other places. However, these results should not overshadow SHS in the workplace environment. SHS exposure in the workplace was the second largest factor influencing cotinine-based SHS exposure rates. Workplaces are a major source of SHS because workers spend the majority of their day at the workplace [21].

In this study, region and house type were the factors influencing smoke exposure; however, the results were slightly different from those of other countries. In other words, unlike studies that reported that rural residents [22] and multiunit housing (MUH) residents [23] are vulnerable to SHS exposure, the present results indicate that the exposure risk of urban [24] and general housing residents [34] was relatively high. To understand these differences, further studies are warranted considering cultural and regional contexts. Concerning residential areas, the status of smoke-free laws for each region may be more important than their rural or urban status [41]. In addition, MUH, such as apartments, may be physically vulnerable to SHS. However, in Korea, it is necessary to consider that housing types reflect socioeconomic status. Apartment residents in Korea tend to have higher socioeconomic levels than residents of other housing types [42].

One of the factors most associated with increasing SHS exposure risk is monthly drinking [25]. In our study, individuals with monthly drinking experiences were 1.17 times more likely to be exposed to SHS. Similarly, Park’s study reported that the higher number of drinking and the amount of alcohol consumed, the higher the concentration of UCo, using the same data source as that used in this study [38]. Drinking behavior causes SHS in various ways. Generally, drinkers have a higher smoking rate [43] and are more likely to be exposed to SHS by smokers who drink with them. Furthermore, drinking places, such as bars, pubs, and night clubs, are more vulnerable to SHS. Thus, the bar is one of the places where smoking status has been difficult to regulate in most countries. Occupational characteristics also influence SHS exposure rates. The results of the present investigation are consistent with previous findings suggesting that individuals with potentially dangerous workplace conditions, including blue-collar workers, have a relatively high exposure risk [21].

### 4.4. Policy Suggestions

Based on our findings, this study suggests the following: first, SHS exposure has disparities. The exposure risks differ depending on sex, age, occupation, region, house type, monthly drinking, and self-reported SHS exposures. Thus, interventions focusing on vulnerable groups should be strengthened. In particular, those who live with smokers in the household and are consequently exposed at home should be a priority intervention group for SHS prevention. Second, biomarkers are the best indicators, reflecting actual body conditions owing to SHS exposure. Therefore, SHS exposure should be assessed using biomarkers. Furthermore, the risk of THS along with SHS has recently been revealed in smoking exposure. However, it is difficult to evaluate THS using SR. In contrast, biomarkers are effective for measuring smoking exposure, including THS. When establishing tobacco control policies, such as the National Health Plan, smoking exposure prevention strategies that reflect cotinine-based measurements are necessary. SR measurement underestimates SHS exposure and erroneously determines the vulnerable groups.

### 4.5. Strength and Limitations

This study has some limitations. First, the study measured the SHS exposure rate by UCo with an acceptable cutoff point that was previously defined. We used a relatively stringent cutoff point (UCo ≤ 20 ng/mL) to exclude, as far as possible, participants who were smokers. However, for some special occupational workers, even nonsmokers may have high UCo. Moreover, generally, the cotinine–creatinine ratio (CCR) can better reflect measuring SHS [44]. In our study, we did not use CCR because some participants did not have creatinine data; however, our study did not aim to find the cutoff point of cotinine. Therefore, consensus on the cutoff point is necessary to consider the use of cotinine-based measurement policy. Nonetheless, these criteria did not affect our findings (Appendix A Figure A1).

Second, because this study was cross-sectional, it found the effects of several factors on SHS exposure but could not explain the causal relationship between each variable and SHS exposure. Third, considering that the KNHANES did not collect information regarding UCo levels between 2012 and 2013, we excluded the data obtained within this period. Unfortunately, considering that only adults were selected, children who are also vulnerable to SHS were not included in this study. Fourth, other factors that may influence UCo concentrations were not considered. For example, there are concerns about potential exposure to nicotine from dietary sources; SHS exposure determined based on urine cotinine content may be compromised [45]. Therefore, we compared several different ranges of SHS exposure using different cutoff points in the Appendix A Figure A1.

Nevertheless, the study used the KNHANES data to ensure the representativeness and reliability of the data. It is meaningful in that it is one of the earliest studies to evaluate SHS using a biomarker, particularly in Asia.

## 5. Conclusions

This study indicates that the actual SHS exposure could not be properly assessed based on SR alone, although the population exposed to SHS by SR had higher UCo concentrations. In particular, SR could not evaluate low-concentration SHS. Official SHS exposure by SRs has declined significantly in the last decade. In contrast, the exposure rate by UCo was similar. Therefore, it is helpful for accurate evaluation to identify low-concentration exposures using biomarkers such as UCo. Considering that even short or low exposure can be harmful to health, the goal of the policy should be to reduce cotinine concentration to keep cotinine concentration as low as possible

## Figures and Tables

**Figure 1 ijerph-18-09284-f001:**
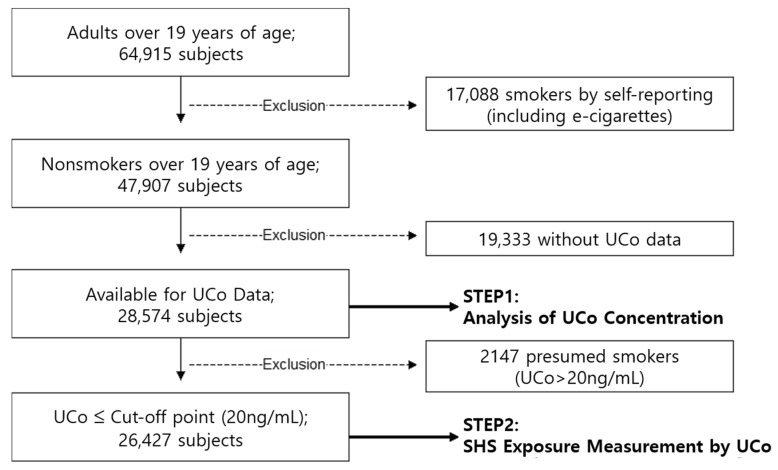
Flowchart of subjects in this study.

**Table 1 ijerph-18-09284-t001:** Urine cotinine concentrations and SHS exposure rate by year.

Measures		Years							
		2009	2010	2011	2014	2015	2016	2017	2018
UCo Concentration ** (Geometric meanand 95% CL, N = 28,574)		9.28 (8.24–10.26)	5.45 (4.85–6.13)	3.88 (3.51–4.27)	1.63 (1.53–1.74)	1.49 (1.39–1.60)	1.03 (0.97–1.10)	1.06 (1.00–1.14)	1.06 (1.00–1.13)
SHS Exposure rate	Self-reporting, at indoor home (No. and weighted%, N = 28,569)	449 (13.4)	148 (13.5)	142 (11.4)	360 (10.2)	321 (7.8)	254 (6.1)	223 (4.6)	173 (3.7)
	Self-reporting, at indoor public places * (No. and weighted%, N = 22,206)	-	-	-	1701 (50.0)	1261 (33.2)	852 (20.7)	808 (19.4)	666 (15.4)
	Self-reporting, at indoor workplaces (No. and weighted%, N = 16,825)	922 (44.0)	424 (53.8)	379 (49.6)	788 (40.9)	617 (28.0)	412 (17.8)	351 (13.6)	337 (12.2)
	Cotinine measurements *** (No. and weighted%, N = 26,427)	2115 (75.1)	1097 (91.4)	1159 (93.1)	3415 (95.9)	3639 (90.2)	3214 (74.9)	3234 (72.6)	3574 (80.0)

* Because entire public facilities were designated as nonsmoking areas in 2012, the SHS exposure rate at indoor public places has been investigated since 2013. ** In 2012–2013, urine cotinine was not investigated, and the survey was resumed from 2014. *** SHS exposure rate: LOD < UCo ≤ 20ng/mL. CL, confidence limit.

**Table 2 ijerph-18-09284-t002:** Urine cotinine concentration in participants by characteristics.

Variables		Participants (N, Weighted %)	UCo (ng/mL)	
			Geometric Mean	95% CL
Sex (N = 28,574)	Male	10,035 (40.4)	2.0	1.9–2.1
	Female	18,539 (59.6)	1.7	1.6–1.8
Age (N = 28,574)	65 over	7485 (17.1)	1.4	1.3–1.5
	45–64	10,945 (37.9)	1.6	1.5–1.7
	19–44	10,144 (45.0)	2.2	2.1–2.3
Income (N = 28,436)	4Q	6651 (26.3)	1.6	1.5–1.8
	3Q	7043 (24.7)	1.7	1.6–1.8
	2Q	7182 (24.9)	1.8	1.7–1.9
	1Q	7560 (26.3)	2.0	1.9–2.2
Occupation (N = 28,574)	Blue-collar	4535 (16.6)	1.8	1.7–1.9
	White-collar	9721 (38.3)	1.8	1.7–1.9
	Agriculture	1520 (3.5)	2.1	1.8–2.4
	Unemployed	12,799 (41.6)	1.8	1.7–1.8
Region (N = 28,574)	Urban	229,992 (84.2)	1.8	1.7–1.8
	Rural	5582 (15.8)	2.0	1.8–2.3
House type (N = 28,574)	General	13,715 (48.3)	2.1	2.0–2.2
	Apartment	14,859 (51.7)	1.6	1.5–1.6
Monthly drinking (N = 28,553)	No	14,709 (46.9)	1.5	1.5–1.6
	Yes	13,844 (53.1)	2.1	2.0–2.2
SHS at home (N = 28,569)	No	26,499 (92.4)	1.7	1.6–1.7
	Yes	2070 (7.6)	4.4	3.9–4.9
SHS at indoor public places (N = 22,206)	No	16,918 (73.2)	1.1	1.1–1.2
	Yes	5288 (26.8)	1.7	1.6–1.9
SHS in indoor workplaces (N = 16,825)	No	12,595 (73.1)	1.4	1.4–1.5
	Yes	4230 (26.9)	3.5	3.2–3.8

**Table 3 ijerph-18-09284-t003:** SHS exposure rate in nonsmokers with identified cotinine levels (unit: number and weighted %).

Variables		LOD < UCo ≤ 20 ng/mL (N = 26,427) *
Overall exposure		21,447 (82.5)
Sex	Male	7789 (86.4)
	Female	13,658 (80.0)
Age	65 over	5493 (78.5)
	45–64	8554 (84.4)
	19–44	7400 (82.5)
Income	4Q	5624 (81.5)
	3Q	5441 (83.2)
	2Q	5304 (81.8)
	1Q	4980 (83.7)
Occupation	Blue-collar	3615 (87.9)
	White-collar	7466 (83.9)
	Agriculture	1103 (81.2)
	Unemployed	9263 (79.2)
Region	Urban	17,399 (82.8)
	Rural	4048 (80.9)
House type	General	10,597 (85.9)
	Apartment	10,850 (79.5)
Monthly drinking	No	10,890 (80.0)
	Yes	10,539 (84.9)
SHS at home	No	19,828 (81.7)
	Yes	1614 (93.0)
SHS in indoor public places	No	12,710 (79.6)
	Yes	4362 (89.3)
SHS in indoor workplaces	No	9599 (82.6)
	Yes	3330 (91.7)

All percentages were weighted to reflect stratified multistage sampling. SHS: secondhand smoke. * Of the 28,574 subjects, those with UCo of >20 ng/mL were excluded from the analysis because they were more likely to be smokers.

**Table 4 ijerph-18-09284-t004:** Factors associated with SHS exposure with identified cotinine levels by multiple logistic regression.

Variables		Univariate	Multiple
		LOD < UCo ≤ 20 ng/mL OR (95% CL)	
Sex	Female (ref) vs. Male	1.58 (1.46–1.72)	1.43 (1.25–1.64)
Age	>65 (ref)	1	1
	45–64	1.30 (1.16–1.44)	1.10 (0.90–1.34)
	19–44	1.48 (1.33–1.64)	1.21 (1.01–1.45)
Income	4Q (ref)	1	1
	3Q	1.02 (0.92–1.14)	0.94 (0.81–1.09)
	2Q	1.12 (0.99–1.27)	1.06 (0.90–1.25)
	1Q	1.17 (1.03–1.33)	1.07 (0.90–1.29)
Occupation	Blue-collar worker (ref)	1	1
	White-collar worker	0.72 (0.63–0.81)	0.77 (0.67–0.90)
	Agriculture	0.60 (0.46–0.78)	0.70 (0.53–0.93)
	Unemployed	0.53 (0.47–0.59)	1.13 (0.85–1.49)
Region	Rural (ref) vs. Urban	1.13 (0.96–1.34)	1.29 (1.04–1.59)
House type	Apartment (ref) vs. General	1.57 (1.40–1.75)	1.55 (1.32–1.81)
Monthly drinking	No (ref) vs. Yes	1.41 (1.30–1.51)	1.17 (1.04–1.32)
SHS at home	No (ref) vs. exposure	2.97 (2.33–3.79)	3.90 (2.56–5.95)
SHS in indoor public places	No (ref) vs. exposure	2.14 (1.90–2.41)	1.51 (1.28–1.79)
SHS in indoor workplaces	No (ref) vs. exposure	2.31 (1.97–2.72)	2.06 (1.64–2.58)

The analysis reflected weights by stratified multistage sampling. CL, confidence limit.

## Data Availability

Data can be downloaded with permission from the Korean CDC KNHANE website (https://knhanes.kdca.go.kr/knhanes/eng) (accessed on 17 June 2021). If you need the processed data, please contact the author to request the data.

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
