# Peer review of "Exposure to Secondhand Smoke: Inconsistency between Self-Response and Urine Cotinine Biomarker Based on Korean National Data during 2009–2018"

_ijerph, 2021, doi:10.3390/ijerph18179284_

Round 1

Reviewer 1 Report

Dear Authors, 

Here are the comments of your paper N°IJERPH-1284601. 

================

The paper IJERPH-1284601 presents the results on exposure to secondhand smoke, the objective was to evaluate the SHS exposure using a biomarker.

Overall, the paper is well written, however, some points are not clear and need further explanations. Several measurements are reported, but there is a lack of any report of how precisely the data was treated and, hence, how the claims are supported. This, from the start, distracts the reader.

Indeed, we suggest describing first the survey and measured parameters, then details of the statistical analysis. Significant rewording is needed. At this stage, a major revision is recommended. Below, the authors may find three specific comments to assist the revision process.

  • The significance and originality of this particular contribution are not explained clearly, and therefore the paper lacks novelty, at least to this reviewer… It is difficult to understand the scientific contribution of the paper. Specifically, the proposal of the authors (new methodology? / new approach? / new equations? / new indicators? / application of an existing methodology for the acquisition of results leading to novel conclusions? / originality lying on the combined application of different techniques?, etc.) and their work's contribution to the existing knowledge is difficult to be specified.

Some important points have to be clarified or fixed before we can proceed, positive action can be taken. Detailed comments are shown in the following.

  1. When explaining technical matters, the paper needs to be much tighter.
    1. My problem with this paper is in the statistical interpretation of data. The results supported by relevant statistical tests (Table 1 and 3). In fact, the comparisons between groups (sex, age, …) in the text do not always seem to follow certain statistical rules. In general, the claims should be supported by specific tests.
    2. What are the most significant factors. That is, searching for the most influential input, or more precisely to rank the input variables from the most influential to the least influential;
    3. There are many other possibilities to analyze these data, a comparison study is necessary.
    4. The detailed description of the measurement campaign and survey is needed to understand the results.
    5. Further discussion on aspects of temporal variability of concentrations is desirable.
  1. I recommend adding the Figure 2 and the interpretation in the results section.
  2. Abstract: The abstract could be improved. The object and the content of the study are not very clearly presented in the abstract text. The description of the applied methodology is rather limited. The abstract should also include the key findings of this research. Currently, it is not very clear. Furthermore,  Good to briefly explain the key results in the abstract.
  3. Some part of the methodology sounds too generic and hard to understand. Good to add some descriptive and example how the method works. Some contents in the methodology subtopics are not sufficient. 
  4. Introduction: The literature review is a little weak. More references on SHS and especially using the data (since it is available under request) should be added. It is better to give more summary and comments rather than just listing them. The research gap and innovation of the paper are not clearly stated in the introduction.

================

Best regards, 

Author Response

Here is a point-by-point response to the reviewers’ comments and concerns.

Based on your comments, this manuscript has been significantly improved. Thank you for your valuable comments.

Comments from Reviewer 1

  • The paper IJERPH-1284601 presents the results on exposure to secondhand smoke, the objective was to evaluate the SHS exposure using a biomarker. Overall, the paper is well written, however, some points are not clear and need further explanations. Several measurements are reported, but there is a lack of any report of how precisely the data was treated and, hence, how the claims are supported. This, from the start, distracts the reader. Indeed, we suggest describing first the survey and measured parameters, then details of the statistical analysis. Significant rewording is needed. At this stage, a major revision is recommended. Below, the authors may find three specific comments to assist the revision process.
  • Response: Thank you for pointing this out. We made a major revision on the material and methods section. Firstly, for understanding the data source, we added a subsection “Korean National Health and Nutrition Examination Survey (KNHANES)”. Then, we revised the way to explain the subject selection process and how to measure each parameter.
  • Comment1: The significance and originality of this particular contribution are not explained clearly, and therefore the paper lacks novelty, at least to this reviewer… It is difficult to understand the scientific contribution of the paper. Specifically, the proposal of the authors (new methodology? / new approach? / new equations? / new indicators? / application of an existing methodology for the acquisition of results leading to novel conclusions? / originality lying on the combined application of different techniques?, etc.) and their work's contribution to the existing knowledge is difficult to be specified.
  • Response: The SHS exposure is commonly measured by self-reporting. This paper aims to point out the limitations of using only self-reporting data as a monitoring indicator in establishing SHS prevention policies.

To this end, firstly, the study suggests positive outcomes of SHS policies by showing a significant decrease in the UCo concentration and the self-reported SHS exposure rate among nonsmokers over the past decade.

Secondly, we suggest that the cotinine measured SHS exposure rate is still high (over 80%) despite the positive changes above, requiring further political effort to reduce it. In particular, the study emphasizes that unrecognized SHS exposure can harm the human body since the cotinine measured SHS exposure was also identified in the groups who reported that they were not exposed to SHS.

Lastly, we would like to confirm the vulnerable groups to SHS exposure in a more precise way by identifying the factors associated with the cotinine measured SHS exposure.

  • We will reinforce the introduction and discussion parts on why this study is important.
  • Comment2: Some important points have to be clarified or fixed before we can proceed, positive action can be taken. Detailed comments are shown in the following. When explaining technical matters, the paper needs to be much tighter.
  • Comment2a: My problem with this paper is in the statistical interpretation of data. The results are supported by relevant statistical tests (Table 1 and 3). In fact, the comparisons between groups (sex, age, …) in the text do not always seem to follow certain statistical rules. In general, the claims should be supported by specific tests.
  • Response: Thanks for your comments. We apologize for not understanding your comments correctly. Does your comment mean statistical significance? If that is the case, it is because, in general, when announcing national statistics by year, rather than statistical verification, only numerical presentations are made. And concerning Table 3, we considered t-test or ANOVA, but in this case, there is overlap with Table 4 (logistic regression), so we did not analyze it.
  • Comment2b: What are the most significant factors. That is, searching for the most influential input, or more precisely to rank the input variables from the most influential to the least influential;
  • Response: As the result of multiple regression, the variables most associated with the cotinine-based SHS exposure were the following: SHS exposure at home (self-reporting), SHS exposure in indoor workplaces (self-reporting), and house type. We will revise the flow of discussion in the order of the most significant variables.
  • Comment2c: There are many other possibilities to analyze these data, a comparison study is necessary.
  • Response: We compared our results with the previous studies conducted using the same data source, and it is described in the discussion section.
  • Regarding the self-reported SHS exposure at home: “According to studies in the US and Spain, which measured SHS exposure by SR and cotinine, self-reported exposure at home has a greater impact on cotinine levels than exposure at work or elsewhere. Similar results are found in the study that assessed the impact of SHS exposure on lung function using the same data source with our study.”
  • Regarding the monthly behavior factor: “Similarly, Park's study reported that the higher number of drinking and the amount of alcohol consumed, the higher the concentration of Uco, using the same data source with this study.”
  • Comment2d: The detailed description of the measurement campaign and survey is needed to understand the results.
  • Response: We described more details on the anti-smoking policies in Korea such as smoke-free law and public campaigns in the discussion section.
  • “In this context, for the past 20 years, South Korea also has committed to protect the public from smoking by gradually expanding smoke-free areas under the National Health Promotion Act, which designated all enclosed places as smoke-free areas in 2015. Public campaigns to raise awareness on the risk of smoking have been implemented through mass media, advertising on buses and subway, information dissemination over the internet, and program to quit smoking. Consequently, smokers are becoming more aware of the risks of SHS exposure and have positively changed their attitudes toward antismoking laws. Additionally, these policies have had a positive effect on creating smoke-free rules in the home.[33].”
  • Comment2e: Further discussion on aspects of temporal variability of concentrations is desirable. a
  • Response: Cotinine is a biologically stable indicator compared to other biomarkers. We further explained the validity of cotinine-based measurement in the introduction and discussion section.
  • Introduction part: “Cotinine is a metabolite of nicotine with a long time half-life of 20-30 hours. Among biomarkers of SHS exposure such as carbon monoxide, carboxyhemoglobin, nicotine, cotinine, tobacco‐specific carcinogen, and carcinogenic DNA abduct, cotinine has been widely used in the clinical field due to its biological stability.”
  • Discussion parts: “Among biomarkers, cotinine is widely acknowledged as a valid indicator of SHS exposure due to its biological stability. It has been used to provide evidence in previous studies which confirmed the causal relationship between SHS and SHS-induced diseases such as lung cancer and cardiovascular disease.”
  • Comment3: I recommend adding the Figure 2 and the interpretation in the results section.
  • Response: We make significant revisions of the result section: The previous "Table 2. the annual change in cotinine concentration and SHS exposure" and "Figure 2. Secondhand smoke exposure rate by year(2009-208)" were brought to the front of the result section together.
  • Comment4: Abstract: The abstract could be improved. The object and the content of the study are not very clearly presented in the abstract text. The description of the applied methodology is rather limited. The abstract should also include the key findings of this research. Currently, it is not very clear. Furthermore, Good to briefly explain the key results in the abstract.
  • Response: Thank you for this suggestion. We corrected the abstract so that it can provide clearer summary of this study.
  • “This study aims to estimate the SHS exposure using urinary cotinine (Uco) to suggest that the SHS exposure could not be properly assessed by self-reporting (SR). Total 28,574 nonsmokers over 19 years old were selected from the KNHANES data for 2009-2018. Firstly, we analyzed the changes in the annual concentration of UCo and measured the annual SHS exposure rates by SR and UCo from 2009 to 2018. Then, the average UCo concentration and the SHS exposure rate measure by UCo were confirmed according to the subjects’ characteristics. Finally, the factors associated with the UCo measured SHS exposure rate was identified by multiple regression analysis. The findings show that the annual Uco concentrations and the self-reported SHS exposure rates have significantly dropped over the past decade. In contrast, the UCo measured SHS exposure indicates that more than 80% of nonsmokers are still exposed to SHS. Moreover, we found the vulnerable groups with the UCo measured SHS exposure. Particular, the self-reported SHS exposure at home and in workplaces and house type were the most associated with the UCo measured. Thus, this study suggests that the actual SHS exposure could not be properly assessed by self-reporting and needs to be verified by a biomarker such as Uco. Considering that even a short exposure can be harmful to health, the goal of the policy should be to reduce cotinine concentration to at least under the LOD level.”
  • Comment5: Some part of the methodology sounds too generic and hard to understand. Good to add some descriptive and examples of how the method works. Some contents in the methodology subtopics are not sufficient. 
  • Response: As mentioned above, we made a major revision to the material and methods section. Firstly, for understanding the data source, we added a subsection “Korean National Health and Nutrition Examination Survey (KNHANES)”. Then, we revised the way to explain the subject selection process and how to measure each parameter.
  • Introduction: The literature review is a little weak. More references on SHS and especially using the data (since it is available under request) should be added. It is better to give more summaries and comments rather than just listing them. The research gap and innovation of the paper are not clearly stated in the introduction.
  • Response: We made a significant revision in the introduction part, including literature reviews on the previous studies conducted in Korea with the same data source.
  • Introduction part: “In this context, several attempts have been made to measure SHS exposure using UCo. Park et al. [301] confirmed the annual trend of SHS exposure using UCo concentrations as an SHS exposure indicator. Jung et al. [302] reported that a certain level of Uco was detected in non-smokers who self-reported with no exposure experience. Moreover, Jeong et al. [33] confirmed that correspondence between the self-reported and cotinine measured SHS exposure rates were low. To sum up, these studies suggest that the actual SHS exposure could not be properly assessed by self-reporting and needs to be verified by a biomarker such as Uco.”

Reviewer 2 Report

The authors study risk factors associated with SHS using public data from 2009 and 2018. They mainly use SCo as subsidiary measure.

I think the paper is poorly structured and that makes that it is very difficult to read and to extract clear conclusions.

First, the authors excluded the smokers (based on self-reporting information) and some additional based on SCo measures. With this criterion, those smokers with SCo below the LOD should not be excluded. I recommend not to exclude patients not self-reporting as smokers. Profiles of smokers participants should be reported (at least as supplementary material).

Next step, to evaluate the concordance between SCo and SR measures. There are statistical procedures for dealing with LOD measures. In any case, you can report that in a separate way.  The SCo levels can be affected for the exposition levels (age and place of exposition).

The analysis of the individual factors are poor and probably affected for trivial confounders. For instance,

“Those who drank more than once a month (monthly 172 drinkers) had higher UCo level (2.1 ng/mL) than those who did not (1.5 ng/mL).”

Perhaps because they spend more time in the bar? Drinking and smoking are frequently related. You are considering non-smokers drinking but the social part could affect that. I this case, you are saying that people we share activities with smokers are SHS (not very surprisingly).

The same with the ‘exposition’, in the non-exposition group we have people no exposes and exposes to other places (not at home but at office) so, to include a variable of type of exposure, for instance, would be helpful – None – Office – Home –Public –More than one…

Introduction: “As a result, adult nonsmokers' exposure to 48 SHS in the workplace decreased from 49.2% in 2010 to 11.9% in 2018, and the exposure 49 rate at home also declined from 14.9% in 2010 to 4.7% in 2018.”

I am not sure that this reduction in CAUSED just for the limitation of smoking in those places. What the smoking rate is? That is, in general, it is a decrement in the number of smokers and that should affect in the number of SHS.

Table 2, the number are a little bit inconsistent. First, third and fifth lines, what did happen between 2011 and 2014?

Based on table 3, 82.5% of people are SHS?

Perhaps more interesting that a univariate analysis (Table 4) would be a multivariate analysis in which the interpretation would be clearer (the confounding here make somethings very difficult to explain)

Author Response

Here is a point-by-point response to the reviewers’ comments and concerns.

Based on your comments, this manuscript has been significantly improved. Thank you for your valuable comments.

Comments from Reviewer 2

  • The authors study risk factors associated with SHS using public data from 2009 and 2018. They mainly use SCo as a subsidiary measure. I think the paper is poorly structured and that makes that it is very difficult to read and to extract clear conclusions.
  • Comment1: First, the authors excluded the smokers (based on self-reporting information) and some additional based on SCo measures. With this criterion, those smokers with SCo below the LOD should not be excluded. I recommend not to exclude patients not self-reporting as smokers. Profiles of smokers participants should be reported (at least as supplementary material).
  • Response: Like SHS exposure measurement, self-reporting has limitations in identifying smokers, so we designed to exclude smokers with cotinine levels once more. Previous studies on smoking or SHS exposure also applied cut-off points to distinguishing smokers and nonsmokers.
  • Please reference the following articles: (1) Kim BJ, Kang JG, Kim JH, et al. Association between Secondhand Smoke Exposure and Hypertension in 106,268 Korean Self-Reported Never-Smokers Verified by Cotinine. J Clin Med. 2019;8(8):1238. (2) Hong JW, Noh JH, Kim DJ. The prevalence of and factors associated with urinary cotinine-verified smoking in Korean adults: The 2008-2011 Korea National Health and Nutrition Examination Survey. PLoS One. 2018;13(6):e0198814.
  • Meanwhile, nonsmokers with cotinine concentrations below LOD are defined as SHS non-exposed groups.
  • Comment2: Next step, to evaluate the concordance between SCo and SR measures. There are statistical procedures for dealing with LOD measures. In any case, you can report that in a separate way.  The SCo levels can be affected by the exposition levels (age and place of exposition).
  • Response: We agree that it is meaningful to confirm the consistency between Self-reported and UCo measured estimates. However, it is out of the scope of this study as it is required other methods and analysis on it; this could be added in the future study.
  • Comment3: The analysis of the individual factors are poor and probably affected for trivial confounders. For instance, “Those who drank more than once a month (monthly 172 drinkers) had higher UCo level (2.1 ng/mL) than those who did not (1.5 ng/mL).” Perhaps because they spend more time in the bar? Drinking and smoking are frequently related. You are considering non-smokers drinking but the social part could affect that. I this case, you are saying that people we share activities with smokers are SHS (not very surprisingly).
  • Response: Drinking Behavior was considered in this study because it increases the likelihood of SHS exposure. Because smoking and drinking are correlated, a person can be easily exposed to SHS from smokers who drink together, and an environment like the bar is also a place with a high risk of exposure to SHS. The fact that the SHS exposure status may vary depending on the drinking behavior may alert smokers to changes in smoking behavior or give evidence to the need for a smoke-free policy.
  • We suggested it in the discussion section. “Drinking behavior causes SHS in various ways. Generally, drinkers have a higher smoking rate and are more likely to be exposed to SHS by smokers who drink with them. Also, drinking places, such as bars, pubs, and nightclubs, are more vulnerable to SHS. Thus, the bar is one of the places where smoking is regulated with difficulty in most countries.”
  • Comment4: The same with the ‘exposition’, in the non-exposition group we have people no exposes and exposes to other places (not at home but at the office) so, to include a variety of types of exposure, for instance, would be helpful – None – Office – Home –Public –More than one…
  • Response: We appreciate your opinion. This paper aims to estimate SHS exposure by UCo to point out that a self-reporting measurement does not properly assess the actual SHS exposure. We agree with your suggestion; however, it is too broad to be considered in this study. Nevertheless, Table 4 indirectly shows which places have a lot of association with the UCo measured estimation. Currently, we are already conducting another research containing your suggestion.
  • Comment5: Introduction: “As a result, adult nonsmokers' exposure to 48 SHS in the workplace decreased from 49.2% in 2010 to 11.9% in 2018, and the exposure 49 rate at home also declined from 14.9% in 2010 to 4.7% in 2018.” I am not sure that this reduction in CAUSED just for the limitation of smoking in those places. What the smoking rate is? That is, in general, it is a decrement in the number of smokers and that should affect in the number of SHS.
  • Response: Among the Articles of WHO FCTC, Article 8 to protect non-smokers from smoking is considered the most effective. A smoke-free law, which restricts smoking in public places, has led to a reduction in the SHS exposure rate with positive changes such as improved public awareness of the risk of smoking, reduced smoking rate, and prevent initial smoking. We described this in detail in the discussion. Response.
  • Comment6: Table 2, the number is a little bit inconsistent. First, third and fifth lines, what did happen between 2011 and 2014?
  • Response: It is because the KNHANES discontinued collecting Uco data in 2012 and 2013 and resumed in 2014. We described it in the subject selection section and the footnotes of the table.
  • Comment7: Based on table 3, 82.5% of people are SHS?
  • Response: This SHS exposure rate (82.5%) is an estimation that includes all subjects for 2009-2018, which differs from the SHS exposure rate of each year (e.g., 80.0% in 2018). We will specify additionally on the table so as not to confuse.
  • Comment8: Perhaps more interesting that a univariate analysis (Table 4) would be a multivariate analysis in which the interpretation would be clearer (the confounding here make somethings very difficult to explain)
  • Response: Thanks for your suggestion. Indeed, we also conducted univariate analyses in the analysis process, and the results were similar to the multiple regression analyses. Although there are some differences, the coefficient direction of univariate and multiple is generally the same. And usually, multiple regression analysis was used because it reduces the error term and provides more information. However, the accompanying univariate analysis may give the reader more information. This is added to table 4. Thank you for your advice.

Round 2

Reviewer 1 Report

Dear, 

Thank you. You have addressed all my concerned.  The paper itself is well written, although somewhat (too) descriptive.  

 The paper can be accepted as is, with minor grammatical corrections. It is recommended that native English speaker conduct a minor revision. 

 The results and Policy suggestions sections can be enhanced. The authors talked about introducing a new cutoff point. This suggestion is a bit naïve, It should provide an "objective" and rigorous standards.
Maybe, the last paragraph (lines 407--409) in the "Policy suggestions" section should be added in the next section, "
Strength and limitations".

It was a pleasure to read this manuscript. I wish the author of the best.

Best regards,  

Author Response

(Response) Thanks for your comments. And concerning your comment, that paragraph (lines 407-409) has also been moved to "Strength and limitations". and added a conclusion section.

Reviewer 2 Report

I think the authors have not made the required changed and that the paper is still poorly presented with no clear conclusions.

Author Response

(Response) I respect your opinion. We have moved some paragraphs of the policy suggestion to the Strength and limitations section. Furthermore, we have asserted the conclusion we want to make through this study in 'Policy Suggestions'. However, we added the 'Conclusion' section because we thought a conclusion was necessary from an academic point of view.

For reference, as you mentioned earlier, ‘those smokers with SCo below the LOD should not be excluded. I feel like our answer is lacking regarding 'I recommend not to exclude patients not self-reporting as smokers.'

We did not exclude participants whose Uco was below LOD. The LOD of UCo is <0.27399 ng/mL, and we excluded subjects above 20 ng/ml as we considered smokers.